# Optimizing Calibration Procedure to Train a Regression-Based Prediction Model of Actively Generated Lumbar Muscle Moments for Exoskeleton Control

**DOI:** 10.3390/s22010087

**Published:** 2021-12-23

**Authors:** Ali Tabasi, Maria Lazzaroni, Niels P. Brouwer, Idsart Kingma, Wietse van Dijk, Michiel P. de Looze, Stefano Toxiri, Jesús Ortiz, Jaap H. van Dieën

**Affiliations:** 1Department of Human Movement Sciences, Faculty of Behavioural and Movement Sciences, Vrije Universiteit, Amsterdam Movement Sciences, 1081BT Amsterdam, The Netherlands; n.p.brouwer@vu.nl (N.P.B.); i.kingma@vu.nl (I.K.); j.van.dieen@vu.nl (J.H.v.D.); 2Department of Advanced Robotics, Istituto Italiano di Tecnologia, 16163 Genova, Italy; maria.lazzaroni@iit.it (M.L.); stefano.toxiri@iit.it (S.T.); jesus.ortiz@iit.it (J.O.); 3TNO, 2316ZL Leiden, The Netherlands; wietse.vandijk@tno.nl (W.v.D.); michiel.delooze@tno.nl (M.P.d.L.)

**Keywords:** back-support exoskeletons, exoskeleton control, load prediction model, optimal calibration

## Abstract

The risk of low-back pain in manual material handling could potentially be reduced by back-support exoskeletons. Preferably, the level of exoskeleton support relates to the required muscular effort, and therefore should be proportional to the moment generated by trunk muscle activities. To this end, a regression-based prediction model of this moment could be implemented in exoskeleton control. Such a model must be calibrated to each user according to subject-specific musculoskeletal properties and lifting technique variability through several calibration tasks. Given that an extensive calibration limits the practical feasibility of implementing this approach in the workspace, we aimed to optimize the calibration for obtaining appropriate predictive accuracy during work-related tasks, i.e., symmetric lifting from the ground, box stacking, lifting from a shelf, and pulling/pushing. The root-mean-square error (RMSE) of prediction for the extensive calibration was 21.9 nm (9% of peak moment) and increased up to 35.0 nm for limited calibrations. The results suggest that a set of three optimally selected calibration trials suffice to approach the extensive calibration accuracy. An optimal calibration set should cover each extreme of the relevant lifting characteristics, i.e., mass lifted, lifting technique, and lifting velocity. The RMSEs for the optimal calibration sets were below 24.8 nm (10% of peak moment), and not substantially different than that of the extensive calibration.

## 1. Introduction

Low back loading during manual material handling in the workplace is a risk factor for low back pain [1,2]. To reduce low back loading during manual material handling, spring-based or actuated back-support exoskeletons can be used to provide support [3]. In actuated exoskeletons, the magnitude of the support is determined by the exoskeleton control strategy and generated by actuated components. Hereto, developers need to decide on the optimal level of desirable support that is required in manual material handling activities [4].

The desirable support can be derived from the biomechanical load on the human body [5,6]. For the low back, a commonly used load parameter is the spinal compression force, which is strongly correlated with the moment imposed on the lumbosacral joint, M_Human_ [7]. Consequently, for back-support exoskeleton control, the desirable support could be determined based on an estimate of M_Human_. These moments are produced by active forces through muscle activation [7], together with passive forces generated through strain of tissues such as muscles, tendons, ligaments, and fascia [8]. Substantial passive forces develop when dorsal tissues are lengthened during forward flexion [8]. If the exoskeleton does not constrain body motion, the passive forces are not affected by the exoskeleton. Consequently, in large flexion angles, the passively generated moment reaches high values [9]. Supporting the full moment (passive + active) would require the user to counteract the exoskeleton to stay in the same posture. Therefore, the desirable support should be derived from the actively generated part of M_Human_ only and the moments generated by passive forces should be excluded.

The active M_Human_ can be predicted with EMG-based trunk muscle models [7,10,11,12,13]. These models must be fitted to each individual through a calibration procedure. The calibration procedure consists of performance of calibration trials during which EMG, kinematics, and external force data are collected. After calibration, these models predict the active M_Human_ using EMG and trunk kinematics data.

The use of multiple surface EMG electrodes will increase preparation time, may be limited by interference with the exoskeleton, and may cause discomfort and thus limit the practical feasibility of EMG-based trunk muscle models in exoskeleton control. To reduce the number of required EMG channels, an additional calibration step has been suggested [14]. This step follows the calibration of the EMG-based trunk muscle model by fitting a regression model on data from the same set of calibration trials. The regression model attempts to find a relation between data from a reduced number of EMG channels together with the kinematic data obtained by sensors embedded in the exoskeleton as predictor variables and the active M_Human_ predicted by an EMG-based trunk muscle model as the response variable. After the calibration and during operation, the regression model provides a real time estimation of the response variable, i.e., active M_Human_, using current values of the predictor variables as input. 

A challenge of this approach is to design an optimal calibration procedure. Previously, EMG-based trunk muscle models and the regression model were calibrated for each user using data obtained during several lifting trials with a range of characteristics, i.e., mass lifted, technique, and velocity [14,15]. However, the time and cost involved with performing an extensive calibration procedure limits practical feasibility as well. In addition, the calibration procedure requires equipment that is not easily implemented in the workplace, such as a motion capture system and force plate. 

This study aimed to determine to what extent the calibration procedure and the number of sensors can be limited while the active M_human_ prediction accuracy remains comparable to that obtained from an extensive calibration procedure. To this end, different sets of calibration trials were employed to calibrate an EMG-based trunk muscle model [7,10] and the regression model. Subsequently, low-back load during symmetric box lifting from the ground and box stacking were predicted using a small number of sensors and the calibrated regression model. Outcomes were compared with a reference. The predictive accuracy was used as a measure of the quality of the regression models achieved by different calibration sets. In addition, the predictive accuracy during lifting from a shelf and pulling and pushing were evaluated, to examine generalizability of the models. In addition, the predictive accuracy obtained when calibrating with quasi static trials, which would allow further simplification of the calibration procedure, was evaluated.

## 2. Materials and Methods

Ten healthy male participants (age: 27.3 ± 2.7 years, weight: 73.8 ± 7.6 kg, height: 1.82 ± 0.09 m) with no history of low-back pain participated in the experiment. 

### 2.1. Informed Consent Statement 

The experimental protocol was approved by the scientific and ethical review board of the faculty of behavioural and movement sciences, Vrije Universiteit Amsterdam (VCWE-2019-086). Participants were informed about the protocol prior to and on the day of the experiment and signed an informed consent.

### 2.2. Data Acquisition 

#### 2.2.1. Body Kinematics, Ground Reaction Force, and Exoskeleton Support

A total of 10 LED cluster markers were attached to the lower legs, upper legs, lower arms, upper arms, pelvis, and trunk (T10 level), as shown in Figure 1a. Feet, hands, and head were considered rigidly attached to the lower legs, lower arms, and trunk, respectively. Using pointer measurements [16], markers were related to anatomical landmarks to construct a 3D linked segment model [17]. Marker positions were collected at a sample rate of 50 Hz using an 3D motion capture camera system (Certus, Optotrak, Northern Digital Inc., Waterloo, ON, Canada, Figure 1b). Moreover, ground reaction forces and moments were measured using two strain gauge based custom-made 1.0 × 1.0 m force plates (Figure 1b) at 200 Hz and resampled to 50 Hz using the “resample” function (MATLAB R2020, The MathWorks Inc., Natick, MA, USA), which applies an FIR antialiasing lowpass filter to the signal and compensates for the delay introduced by the filter. Subsequently, marker positions and force-plate data were low-pass filtered with a 5 Hz cutoff frequency using a second-order Butterworth filter to avoid noise amplification in further data processing stages and synchronized using a trigger signal generated by the motion capture system. In addition, during the trials with the exoskeleton, the assistive moment generated by the exoskeleton was measured by embedded torque sensors and stored as M_Exo_. Exoskeleton data were synchronized with the positional marker data using the time delay between the trunk rotation angle calculated from the marker positions and the trunk rotation angle measured by the exoskeleton embedded sensor. The time delay was determined based on the cross-correlation analysis between these two signals.

#### 2.2.2. Muscle Activity

Twelve pairs of Ag/AgCl surface EMG electrodes (BlueSensor N, Ambu A/S, Ballerup, Denmark) with inter-electrode distance of approximately 20 mm were attached to the shaved and cleaned skin (Figure 1c,d). The electrodes were bilaterally placed over rectus abdominis (ventrally at the umbilicus level), internal oblique (superior to the inguinal ligament), external oblique (mid-axillary line, halfway between the iliac crest and the lowest edge of the ribcage), Iliocostalis lumborum (6 cm lateral to L2), longissimus thoracis pars lumborum (3 cm lateral to L1), and pars thoracis (4 cm lateral to T9), as described by [18]. EMG signals were amplified (Porti-17TM, TMSi, Enschede, The Netherlands) and stored at 2000 Hz. Off line, EMG signals were band-pass filtered (10–400 Hz) with a second-order Butterworth filter, filtered to remove the electrical noise [19], high-pass filtered (30 Hz) to remove ECG artifacts [20], full-wave rectified, and low-pass filtered with a cut-off frequency of 2.5 Hz [21] to determine the linear envelope of the signal [22]. The signals were normalized to the maximal voluntary contractions (MVC), synchronized with kinematics data and resampled to 50 Hz.

### 2.3. Experimental Design and Procedure 

Initially, participants performed maximum exertion tasks to obtain MVC of the trunk muscles as described by [23]. In short, they activated their back and abdominal muscles with a maximum effort. The peak values of the linear envelope of the EMG signals were defined as the MVC for each muscle. Then, they carried out different work-related tasks, namely, (A) lifting from the ground without an exoskeleton, (B) lifting from the ground with an exoskeleton, (C) box stacking, (D) lifting from a shelf, and (E) pulling and pushing (Figure 2). All trials were performed with two repetitions.

During A, participants lifted a box from the ground to the upright posture, followed by placing it back on the ground. Data obtained from these lifts were used for the calibration as explained in the next section and will be referred to as calibration trials. These lifts were performed in 14 (2 × 2 × 2 + 2 × 1 × 3) conditions with different lifting characteristics, i.e., mass lifted (7.5 and 15 kg), lifting technique (stoop (keeping knees extended and reaching the box with trunk flexion) and squat (keeping trunk extended and reaching the box with knee flexion), Figure 3), and lifting velocity (normal and slow). In addition, they lifted the same boxes using a free lifting technique (a self-selected combination of knee and trunk flexion) at very slow, normal and fast velocities. 

During B, participants performed the box lifting task as (A). They lifted 7.5 kg and 15 kg boxes in a free lifting technique and with normal and fast velocity while wearing an actuated back-support exoskeleton, XoTrunk (INAIL/Italian Institute of Technology (IIT)) [24]. The lifts were repeated with different control strategies governing the exoskeleton support, namely, INCLINATION (assistive torque adjusted proportional to trunk inclination angle), DYNAMIC (assistive torque adjusted proportional to trunk inclination angle and angular acceleration), VELOCITY (assistive torque adjusted proportional to trunk inclination angle and angular velocity), HYBRID (assistive torque adjusted proportional to trunk inclination angle and forearm muscle activity), and TRANSPARENT (assistive torque compensates the friction and inertia of the motors, so that the user perceives the exoskeleton support as in *zero-torque* mode). 

During C, participants unstacked and re-stacked a pile of three boxes. Participants lifted one box at a time, carried it for approximately two meters and stacked it onto a new pile from the ground up. Next, they unstacked the two top boxes from the new pile by lifting the second box from the top (so they carried two boxes at a time), carried them for approximately two meters and placed them on the ground. Each box weighed 7.5 kg. These trials were performed while wearing the exoskeleton controlled by the INCLINATION strategy. 

During D, participants held a box at hip level in the upright posture, then placed it on the top of a surface at 1.5 m height, followed by lowering it back to hip level in the upright posture. They performed these trials without the exoskeleton and with 7.5 kg and 15 kg boxes. 

During E, participants performed isometric pulls and pushes against a handle that was attached to a rope. The rope passed over a pulley and was tensioned by a hanging mass of 10 kg. These trials were performed without the exoskeleton. The choice of performing D and E without the exoskeleton was based on the fact that they were performed in largely upright posture, while the exoskeleton’s support was based on trunk inclination angle. 

Data obtained during B, C, D, and E (all together referred to as the test trials) were used to evaluate the predictive accuracy of the models.

### 2.4. Calibration Sets

Different calibration sets were used to calibrate the EMG-based muscle model (EMGMod) and subsequently, train the regression model (RegMod). One calibration set (referred to as the Full set) included all calibration trials (the ones performed during A, i.e., lifting from the ground without the exoskeleton in 14 conditions with different mass lifted, lifting technique, and velocity, number of trials *n* = 14). The predictive accuracy achieved by the Full set was considered as the reference predictive accuracy and was used as the first calibration set for our final analysis. 

Additionally, any subsets of Full set, with *n* (*n* ∈ {1, 2, 3, …}) number of calibration trials included in the set, were generated and used for the calibration. The value of *n* was increased until the predictive accuracy of RegMod approached that of the Full set. Therefore, the final *n* value, defined as R, represents the minimum number of calibration trials required to obtain the reference quality. Given that increasing the variability of data in the calibration set may improve the performance of the models [25], R-element calibration sets, which include trials that cover both extremes of the three lifting characteristics, i.e., mass lifted, lifting technique, and lifting speed, were defined as Selected family of calibration sets.

Among the Selected family, the calibration set with the poorest accuracy was defined as the Worst-selected, which was used as our second calibration set for our final analysis. The predictive accuracy obtained with this set helped us determine whether the selection criterion is sufficient to achieve a comparable predictive accuracy as the reference predictive accuracy. 

Finally, as a third calibration set for our final analysis, a set of only the trials with the free lifting technique at a very slow speed and with 7.5 kg and 15 kg boxes was defined. This set is denoted as the Quasi-static set. During these trials, dynamically-induced loads are limited; therefore, calibration can be conducted in practice with a limited motion capture system and force gauges. 

### 2.5. Data Processing

#### 2.5.1. Pipeline

The predictive accuracy of each calibration set was determined following the data processing pipeline shown in Figure 4 for each calibration set. In the pipeline, three models—inverse dynamics (ID), EMGMod, and RegMod—were used, each described in detail in the next section. First, ID was implemented to predict the moment around the lumbar spine, $M_{\mathrm{Human}}^{\mathrm{ID}}$, for all trials. The superscript denotes the model used for the estimation. As ID provides accurate prediction, $M_{\mathrm{Human}}^{\mathrm{ID}}$ was considered the reference for $M_{\mathrm{Human}}$. Second, the calibration procedure was performed for each individual using calibration set data. More specifically, initially the EMGMod was calibrated using data from 12 EMG channels together with ID outcomes to differentiate between active and passive components of M_Human_. Next, the RegMod was trained using data from four EMG channels (bilateral longissimus thoracis pars lumborum and pars thoracis) together with pelvic and trunk kinematics as predictor variables and the active moment, predicted using the calibrated EMGMod, as the response variable. The selection of these specific EMG channels was shown to provide the best performance of the regression model [14]. Third, the calibrated EMGMod and RegMod were applied on the test trials to predict the active moment, i.e., $M_{\mathrm{Active}}^{\mathrm{EMGMod}}$ and $M_{\mathrm{Active}}^{\mathrm{RegMod}}$, respectively. In this step, i.e., implementing calibrated models, EMGMod required data from 12 EMG channels whereas RegMod required four EMG channels. 

The active moment is the intended factor for exoskeleton control, but an appropriate reference value is unavailable. Therefore, the accuracy of $M_{\mathrm{Human}}$ prediction was used to determine model quality. To evaluate the accuracy of $M_{\mathrm{Human}}$ prediction corresponding to each calibration set, the moment that was generated by passive forces and predicted with EMGMod, $M_{\mathrm{Passive}}^{\mathrm{EMGMod}}$, was added to $M_{\mathrm{Active}}^{\mathrm{EMGMod}}$ and $M_{\mathrm{Active}}^{\mathrm{RegMod}}$ to predict $M_{\mathrm{Human}}^{\mathrm{EMGMod}}$ and $M_{\mathrm{Human}}^{\mathrm{RegMod}}$, respectively. Note that $M_{\mathrm{Passive}}^{\mathrm{EMGMod}}$ was used for evaluation purposes only and it is not needed for the operation phase. Finally, root mean square errors (RMSE) between $M_{\mathrm{Human}}^{\mathrm{ID}}$ and $M_{\mathrm{Human}}^{\mathrm{EMGMod}}$ and between $M_{\mathrm{Human}}^{\mathrm{ID}}$ and $M_{\mathrm{Human}}^{\mathrm{RegMod}}$ were determined and used as a measure of predictive accuracy for EMGMod and RegMod, respectively. It should be noted that, as RegMod is calibrated on the basis of EMGMod outcomes, RegMod outcomes are subject to the errors of EMGMod. Therefore, RegMod is expected to perform worse than EMGMod. However, given the smaller number of required EMG sensors, implementing RegMod is practically desired. 

#### 2.5.2. Evaluation and Statistical Analysis

To evaluate the influence of calibration sets on the performance, RMSEs of all test trials were calculated, averaged across participants, and assigned to the corresponding calibration set.

In addition, to assess the impact of different calibration sets on the models’ performance for tasks specifically, test trials were grouped in task groups, i.e., lifting from the ground, box stacking (carrying and lifting phase separately), lifting from a shelf, and pulling and pushing, and RMSEs were calculated. Repeated measures ANOVA followed by a post-hoc Bonferroni test were performed on RMSEs to determine the main effects and interactions of task group (5 levels: B, carrying phase of C, lifting phase of C, D, and E) and calibration set (3 levels: Full, Worst-selected, and Quasi-static). A significance level of 0.05 was used for all tests.

### 2.6. Models

#### 2.6.1. Inverse Dynamics

Using lower body kinematics and force plate data, the net L5/S1 joint moment, $M_{\mathrm{Net}}^{\mathrm{ID}}$, was calculated using a bottom-up inverse dynamics model [17]. In the trials without the exoskeleton, $M_{\mathrm{Net}}^{\mathrm{ID}}$ equals the $M_{\mathrm{Human}}^{\mathrm{ID}}$. In the trials with the exoskeleton, $M_{\mathrm{Net}}^{\mathrm{ID}}$ is carried by the human and the exoskeleton ($M_{\mathrm{Human}}^{\mathrm{ID}}=M_{\mathrm{Net}}^{\mathrm{ID}}-M_{\mathrm{Exo}}$).

#### 2.6.2. EMG-Driven Muscle Model (EMGMod)

Using the EMGMod, $M_{\mathrm{Active}}^{\mathrm{EMGMod}}$ and $M_{\mathrm{Passive}}^{\mathrm{EMGMod}}$ were predicted using trunk muscle EMG signals, lumbar kinematics, and seven parameters (P), which represent trunk-muscle contractile properties [7,10]. The parameters (P) were calibrated to each individual to create a subject-specific model. 

The parameters (P) include a scaling factor between EMG signal amplitude and muscle stress, a position of the passive length-tension curve relative to the optimum muscle length, a scaling factor for the passive length-tension curve in the model, two scaling factors for the eccentric part and the concentric part of the active tension-velocity curve, and the optimum angle, defined as the flexion angle at which trunk muscles are at the optimum length [7]. During EMGMod calibration, the objective function J (as described in Equation (1)) was minimized using the interior-point method of the nonlinear programming solver “fmincon” (MATLAB R2020, The MathWorks Inc., Natick, MA, USA) to determine the appropriate P for each individual.
(1)$$J=\int_{0}^{T}{\left(M_{\mathrm{Human}}^{\mathrm{ID}}-{\;M}_{\mathrm{Active}}^{\mathrm{EMGMod}}\left(P\right)-M_{\mathrm{Passive}}^{\mathrm{EMGMod}}\left(P\right)\right)}^{2}\mathrm{dt}$$

#### 2.6.3. Regression Model (RegMod)

RegMod predicted $M_{\mathrm{Active}}^{\mathrm{RegMod}}$ using EMG data from two bilateral back muscles, i.e., longissimus thoracis pars lumborum and pars thoracis, together with kinematic data, which can be obtained from the sensors embedded in the exoskeleton, i.e., trunk inclination angle, angular velocity and angular acceleration (the rotation of thorax with respect to the global z-axis), trunk flexion angle (the angle of the thorax relative to the thigh), and hip angle (the angle of the pelvis relative to the thigh) [14]. 

Different types of regression models were used from the Regression Learner App (MATLAB R2020, The MathWorks, Natick, MA, USA). The Coarse Gaussian SVM model yielded the highest accuracy and was therefore used in this study. Moreover, 10-fold cross-validation was used to prevent overfitting [26]. The regression model was trained for each participant using the calibration set data and subsequently implemented to predict $M_{\mathrm{Active}}^{\mathrm{RegMod}}$.

## 3. Results

The predictive accuracy for EMGMod and RegMod varied for different subjects, trials, and calibration sets. Examples of $M_{\mathrm{Human}}$ prediction for a lifting trial with good, average, and poor predictive accuracy are shown in Figure 5. 

Averaged-across-participants RMSEs of EMGMod over all test trials ranged between 20.3 and 25.9 nm and those of RegMod ranged between 21.8 and 34.8 nm for any combination of one to three calibration trials. RMSEs depended on the number of lifting trials included in the calibration sets and on the characteristics of the included trials, as shown in Figure 6. 

For EMGMod, the Full calibration accuracy level was nearly achieved using a calibration set including only one trial. In contrast, for RegMod, the Full calibration accuracy level was achieved by the best set of calibration sets with three trials included. Therefore, the Selected sets were chosen from calibration sets with *n* = 3 trials included. For Selected sets, i.e., the sets maximizing variation in lifting characteristics, the RMSEs of EMGMod ranged between 20.3 and 22.0 nm and those of RegMod ranged between 21.8 and 24.8 nm. From the Selected sets, the one with the largest RMSE deviation from the Full calibration level was defined as the Worst-selected set.

Averaged over all tasks, the RMSEs for Quasi-static, Worst-selected, and Full calibration sets were, respectively, 22.4 nm, 22.0 nm, and 20.7 nm for EMGMod and 29.7 nm, 24.8 nm, and 21.9 nm for RegMod. 

Surprisingly, the effects of calibration set (*p* = 0.39) and interaction between task group and calibration set (*p* = 0.30) were not significant for EMGMod (Figure 7). However, a significant effect of task group (*p* = 0.01) was found for EMGMod. Pairwise comparison Bonferroni tests showed that RMSEs for pulling/pushing tasks were lower than those for lifting from the ground (*p* = 0.01), lifting phase of the stacking task (*p* = 0.01), and lifting from a shelf (*p* = 0.04). Moreover, for the stacking task, the RMSEs of the carrying phase were smaller than those of the lifting phase (*p* = 0.03). 

For RegMod (Figure 7), the main effect of task group (*p* < 0.01) was significant but that of calibration set (*p* = 0.18) was not significant. However, in contrast to EMGMod, a significant interaction was found between task and calibration set (*p* = 0.01). Within each task group, a univariate repeated measures ANOVA followed by a post-hoc Bonferroni testing were performed. These tests indicated that the RMSEs of the Worst-selected set were not significantly different from those of the Full set for all task groups. In contrast, RMSEs of the Quasi-static set were larger than those of the Full set for lifting from the ground (*p* < 0.01) and for the lifting phase of the stacking task (*p* = 0.04). For the carrying phase of the stacking task, lifting from a shelf, and pulling and pushing tasks, the differences between Quasi-static and Full were not significant.

## 4. Discussion

This study aimed to assess the effects of using different sets of model calibration trials on the accuracy of predicting the moment generated actively by the human, active $M_{\mathrm{Human}}$. To predict the active $M_{\mathrm{Human}}$, three models, i.e., an inverse dynamic model (ID), an EMG-based trunk muscle model (EMGMod), and a regression model (RegMod), were used in two phases of the data analysis, i.e., calibration and operation. In the calibration phase, the biomechanical data obtained during each set of calibration trials was used to solve the ID, calibrate the EMGMod, and train the RegMod, successively. In the operation phase, the RegMod was used to predict the active $M_{\mathrm{Human}}$ during a set of test trials. The predictive accuracy of each set of calibration trials was calculated by comparing $M_{\mathrm{Human}}$ predicted with RegMod with $M_{\mathrm{Human}}$ predicted with the ID during the test trials. 

RMSEs were used to quantify prediction errors [27] and to assess the accuracy achieved by different calibration sets. In addition, calibration sets were compared using the coefficient of determination (R^2^). The results of R^2^-based comparison agreed with the results based on the RMSEs. Given that RMSE is an absolute measure of the fit whereas R^2^ is a relative measure of the fit, RMSE suits the aim of implementing the model in exoskeleton control better. Therefore, it was selected for the evaluation and statistical analysis. 

The results suggest that the predictive accuracy of the RegMod depends not only on the goodness of fit during the calibration procedure, but also on the accuracy of the response variable, i.e., active $M_{\mathrm{Human}}$ predicted with EMGMod. Given that using different sets of calibration trials affect the outcome of EMGMod, $M_{\mathrm{Human}}$ predicted with EMGMod was compared with $M_{\mathrm{Human}}$ predicted with the ID, in order investigate the sources of errors in further detail. 

The results indicated that the overall RMSEs attained by an extensive calibration procedure, the Full calibration set, were 20.3 nm for the EMGMod and 21.8 nm for RegMod. The relatively small difference between RMSEs of EMGMod and RegMod suggests that using the Full calibration set leads to minor errors caused by RegMod fitting.

### 4.1. Calibration Sets 

From these results, it is clear that both models can obtain close to maximal predictive accuracy when a properly limited set of calibration trials was employed.

The results suggest that for EMGMod, using one calibration trial may suffice to adjust the subject-specific parameters. This may be because, unlike the RegMod, the EMGMod is established based on biomechanical principles of the musculoskeletal system and the calibration is required for adjusting subject-specific parameters of the model. Consequently, very few calibration trials may provide enough information to approach the maximal accuracy. However, calibration through three trials based on the Selected Sets would provide more robust accuracy. 

For RegMod, the results indicate that a set of three calibration trials are required to approach maximal predictive accuracy. The RMSEs of the Selected sets (ranged from 21.8 and 24.8 nm) were all within a close range or equal to the Full calibration RMSE of RegMod (21.8 nm). In addition, no significant difference was found between the results of the Worst-selected set and the Full calibration. This suggests that the selection criterion, i.e., using each extreme of the lifting characteristic, ensures a calibration set that leads to a close to maximal predictive accuracy. The reason might be that the selection criterion results in a combination of trials together covering a sufficient range of input variables of the RegMod. 

### 4.2. Model Performance for Different Task Groups

For EMGMod, employing the Quasi-static, Worst-selected, or Full calibration set did not significantly affect the RMSEs. This is in line with a few calibration trials being sufficient for EMGMod calibration (as described above). Regarding the significantly lower RMSEs of pulling/pushing compared to lifting tasks, it should be noted that pulling/pushing tasks were performed quasi-static and in the mid-range of trunk muscle length. Therefore, these tasks are less affected by errors in the modeling of muscle force-velocity and force-length relationships. In addition, the lumbar spine moments are usually lower during pulling/pushing than lifting tasks [17,28,29]. Therefore, relatively lower RMSEs are expected during pulling/pushing, given that the model prediction errors are expected to be proportional to the absolute moment values. The same reasoning holds true for lower RMSEs of the carrying phase compared to the lifting phase of the stacking task, which is quasi-static for trunk muscles and causes relatively lower lumbar spine moments.

For RegMod, RMSEs of lifting from the ground and the lifting phase of the stacking task were significantly lower when calibrating with the Worst-selected or the Full sets compared to the Quasi-static set. In contrast, RMSEs of the other task groups were not affected by calibration set. Clearly, the Quasi-static set insufficiently covered dynamics for RegMod calibration, and this mainly deteriorated predictions in dynamic lifting tasks.

### 4.3. Limitations

One major limitation of the current study is the errors generated by the EMGMod. The results indicate that the accuracy of the RegMod is limited by the performance of the EMGMod, as errors were comparable between RegMod and EMGMod. Therefore, enhancing EMGMod performance would help to improve the accuracy of RegMod. In such a case, the proposed selection criterion for the calibration set may not ensure reaching the maximal accuracy. Another limitation is that the reported predictive accuracies corresponding to different calibration sets were calculated based on the total $M_{\mathrm{Human}}$ prediction, whereas the actively generated part of M_Human_ is the intended outcome for exoskeleton control. This was due to the unavailability of an appropriate reference value for the active moment. The reported RMSEs might differ from the error in active $M_{\mathrm{Human}}$ prediction. Nevertheless, an optimal prediction of active $M_{\mathrm{Human}}$ depends on an adequate prediction of total $M_{\mathrm{Human}}$.

Another limitation of this study is the repeatability of the results. The results of the statistical analysis indicate that the conclusions hold true for the young healthy male population. In addition, given two repetitions of each experimental trial, minor kinematic variations in performing the same task were incorporated in the dataset. RegMod fit and parameters may change over time considering the changes in predictor variables that may occur, for instance due to replacement of EMG electrodes, or due to fatigue-related changes in EMG signals. Note that EMG signal normalization will limit the electrode replacement effects as the normalized EMG activity is less prone to changes caused by electrode replacement. It should be considered that to implement the model in exoskeleton control, the predictive accuracy should be sufficiently robust, and re-calibration procedures may be needed. The predictive accuracy for prolonged working hours, its consistency over days, and its changes due to fatigue should be investigated in future research.

Finally, participants in this study were all males, with similar age, weight, and height and with no specific manual material handling experience. The reason was that the size of the available exoskeleton constrained the height and weight of the participants. So only males with similar body dimensions were recruited. In addition, older adults and experienced workers were excluded in view of limited availability. Results may differ for the excluded groups due to anatomical differences and differences in lifting behavior. More diverse subject groups should be included in future research to examine the generalizability of the conclusions.

### 4.4. Implications for Practice

Given the objective to implement the model in exoskeleton control, the required sensors for the calibration phase and for the operation phase should be considered. For calibration, the results indicated that the maximal predictive accuracy was approached with either of the Selected sets. These sets were obtained with high-end motion capture systems and force gauges, which typically are available in the laboratory settings. This may limit the practical feasibility of implementing this approach. To prevent this, the Quasi-static calibration set, which potentially can be obtained with a limited motion capture systems and force gauges, was introduced. However, this set was not sufficient to obtain close to maximal predictive accuracy. Alternatively, ambulatory measurement systems [30] may be used for calibration to solve this issue.

For the operation phase, the number of sensors was already reduced by using the RegMod approach. However, this approach still requires four EMG signals that can be obtained by ambulatory devices for EMG monitoring. Note that during the operation phase, the (close to) real-time processed EMG signals together with kinematics data captured with sensors embedded in the exoskeleton should be used as the input of RegMod to predict the current active $M_{\mathrm{Human}}$ and control the exoskeleton accordingly.

## 5. Conclusions

The present study shows that in order to train a regression-based prediction model of active $M_{\mathrm{Human}}$, limited sets of calibration trials are sufficient to obtain close to maximal predictive accuracy for symmetric lifting from the ground. The results suggest a criterion to optimally design a limited set; it should consist of three lifting trials such that each extreme of three relevant lifting characteristics, i.e., mass lifted, lifting technique, and lifting velocity, are covered. An optimal calibration procedure reduces time and effort required and, consequently, improves the practical feasibility of implementing this regression-based prediction model in applications such as active exoskeleton control.

## Figures and Tables

**Figure 1 sensors-22-00087-f001:**
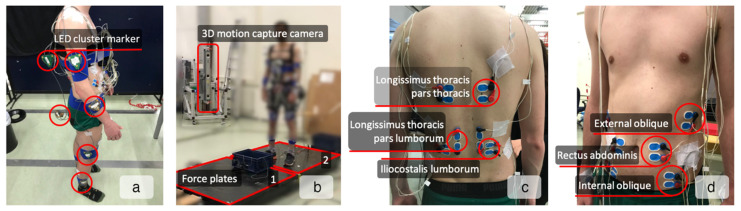
Experimental setup. (**a**) Ten LED cluster markers were attached to lower legs, upper legs, lower arms, upper arms, pelvis, and trunk (T10 level). The cluster markers on the pelvis and trunk were attached using rigid structures for better visibility. The cluster markers attached to the left lower leg, left upper leg, left lower arm, and left upper arm are not visible due to the camera’s position. (**b**) Three-dimensional motion capture cameras around the experimental setup collected marker positions and force plates measured ground reaction forces and moments. One out of four cameras is visible. (**c**,**d**) Twelve pairs of EMG electrodes on the back (**c**) and abdominal (**d**) trunk muscles.

**Figure 2 sensors-22-00087-f002:**
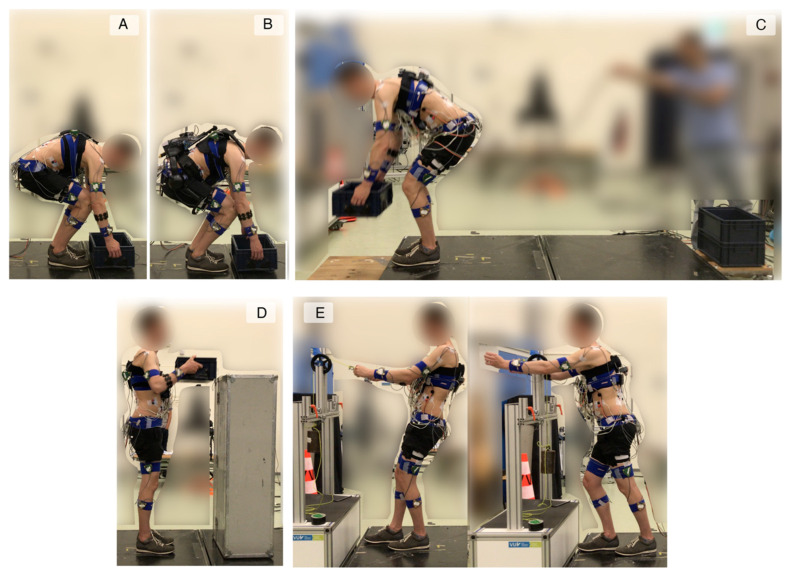
Participants performed different work-related tasks: (**A**) lifting from the ground without an exoskeleton, (**B)** lifting from the ground with an exoskeleton, (**C**) box stacking, (**D**) lifting from a shelf, and (**E**) pulling and pushing.

**Figure 3 sensors-22-00087-f003:**
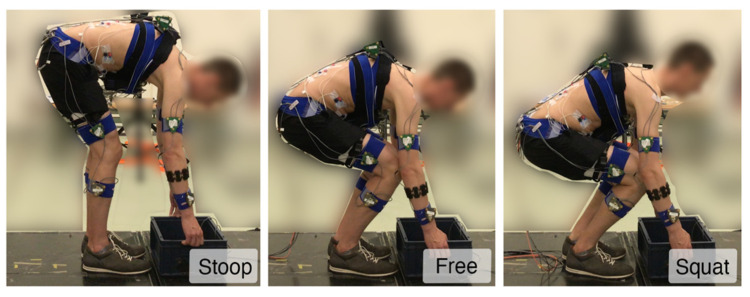
Participants lifted the box with different techniques: (**Stoop**), keeping knees extended and reaching the box with trunk flexion; (**Free**), a self-selected combination of knee and trunk flexion; (**Squat**), keeping trunk extended and reaching the box with knee flexion.

**Figure 4 sensors-22-00087-f004:**
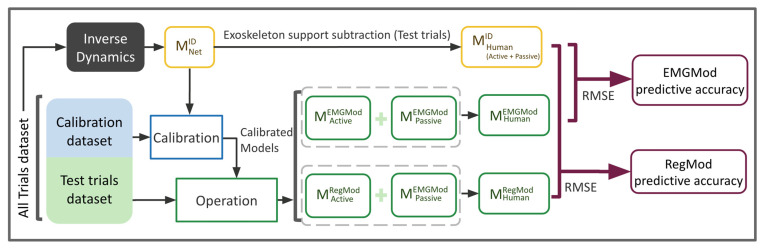
Data processing and evaluation pipeline for a calibration set. Inverse dynamics (ID) was implemented to determine the moment around the lumbar spine ($M_{\mathrm{Human}}$) for all trials. Next, the dataset collected during a set of calibration trials was used to calibrate the EMG-driven muscle model (EMGMod) and the regression model (RegMod), successively. Calibrated models were used to predict the active $M_{\mathrm{Human}}$ during a set of test trials. To evaluate the predictive accuracy achieved by the calibration set, the total $M_{\mathrm{Human}}$, determined by ID, was compared with that predicted by the EMGMod and that predicted by the RegMod.

**Figure 5 sensors-22-00087-f005:**
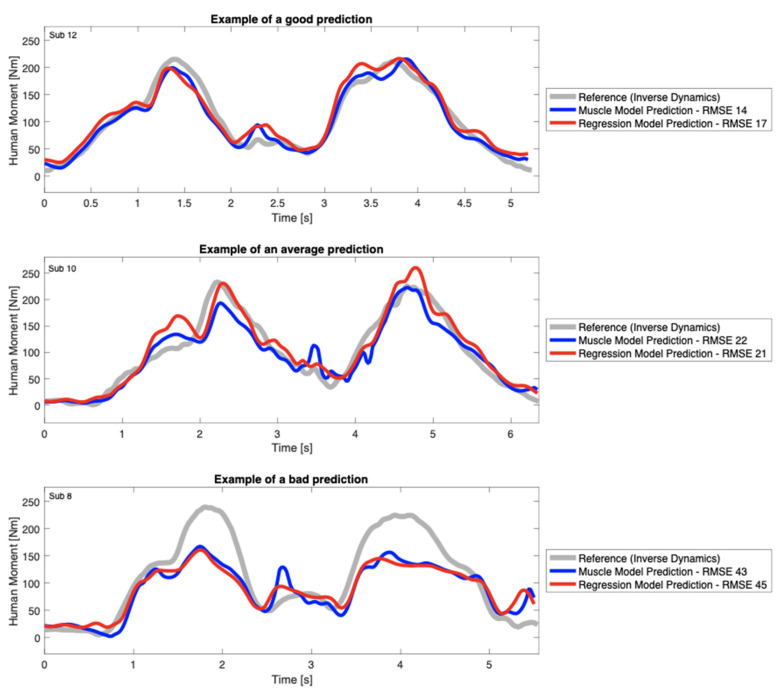
Examples of lumbar spine moment prediction during symmetric box lifting task. The reference values were determined by inverse dynamics, and the predictions were made by EMGMod and RegMod. The predictive accuracy for EMGMod and RegMod varied for different subjects, trials, and calibration sets. These examples were selected to illustrate the inverse relationship between the predictive quality and the RMSE of the models.

**Figure 6 sensors-22-00087-f006:**
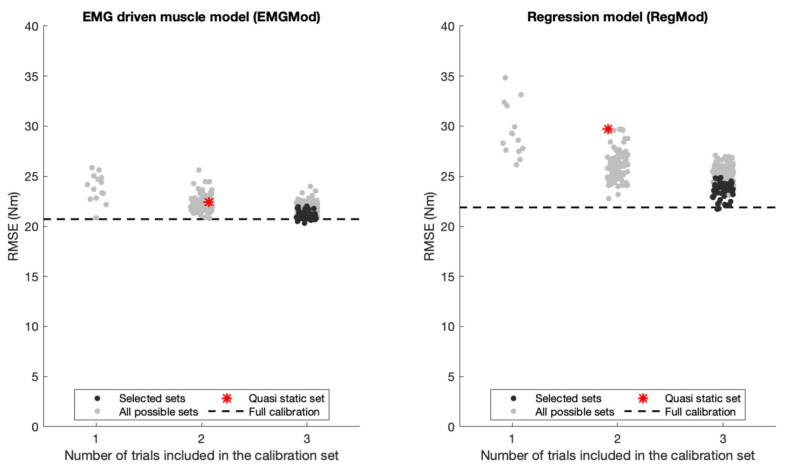
Averaged across participants RMSEs of EMGMod and RegMod for different calibration sets. RMSEs depended on the number of lifting trials included in the calibration sets and on the characteristics of the included trials. The dashed line represents the reference RMSE that was attained by Full calibration. The number of trials included in the calibration sets was increased until the RMSE of RegMod approached the reference RMSE. The family of Selected sets, which was chosen based on the selection criterion, resulted in RMSEs within a close range of the reference RMSE. The Quasi static set resulted in larger RMSE than Full set.

**Figure 7 sensors-22-00087-f007:**
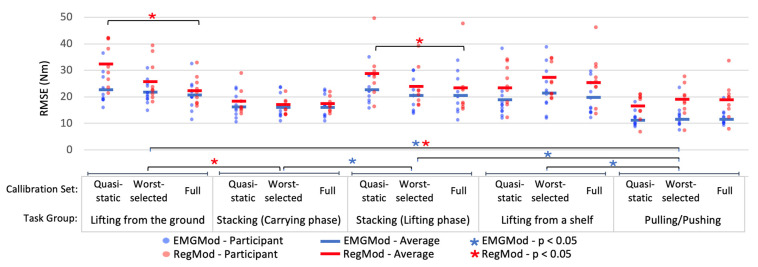
Task-specific RMSEs of EMGMod and RegMod for Quasi-static, Worst-selected, and Full calibration sets. RMSEs of Worst-selected calibration set were not significantly different from those of Full set for all task groups. Quasi-static set resulted in larger RMSEs than Full set for lifting from the ground and the lifting phase of the stacking task.

## Data Availability

The data that support the findings of this study are available from the corresponding author, A.T., upon reasonable request.

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
