# Peer review of "Optimizing Calibration Procedure to Train a Regression-Based Prediction Model of Actively Generated Lumbar Muscle Moments for Exoskeleton Control"

_sensors, 2021, doi:10.3390/s22010087_

Round 1
Reviewer 1 Report
This paper provides us an overall view about the optimization model for muscle moments evaluation. Basically, it demonstrates the entire project model frame which covers the data acquisition, model evaluation and result analysis. The paper can be considered for acceptance if the authors can address the following points: 1. In the Data acquisition session, more figures could be included to show how the sensors are applied to get data. Especially in ‘2.1.1 Body Kinematics, ground reaction force and exoskeleton support’ and ‘2.1.2. Muscle activity’, the data acquisition methods can be delivered more clearly. 2. How is the repeatability of the experiment results? And whether the limited sample numbers would ensure the reliability of the conclusion? It seems there is a lack of discussion and support arguments concerning the above points. 3. In 2.4.2 Evaluation and Statistical Analysis, more factors/parameters could be included in the evaluation of the model. The authors should also include more discussion on the comparison of different models. 4. The participants mainly fall within one group with similar age, weight and height. More subject groups with different distribution should be included to make it more convincing.Author Response
Dear Reviewer 1,
Thank you for your review and fruitful comments and suggestions. The manuscript was revised according to the reviewers’ comments. Please see the attachment for the authors’ responses to your comments and the details of the revisions to the manuscript.
Bests regards,
Ali Tabasi
Corresponding author

Reviewer 2 Report
The present manuscript describes an optimization calibration procedure intended to train a regression-based prediction model for human-related lumbar movements attached or not to an exoskeleton. To that end, different lumbar-related work-tasks were deployed, as well as control strategies for exoskeleton operation on a single human trial divided into optimization (calibration) and operation stages. The authors then used the RMSE metric to compare between extensive and limited calibration sets. Although the human trial was largely well-conducted by the authors, some technical questions raised my attention regarding the description of the applied method and/or possible improvement suggestions for the organization of the manuscript itself, as follows:
1) Abstract: it is said that the RMSEs for the optimal calibration sets were below 24.8 Nm. Does this value represent the state-of-the-art for this field? Is there any biological substantiation?
2) Lines 105 – 111: more technical details about the force plates used during the trial are needed in the manuscript as they were “custom-made” by the authors. Which type of force sensor was used? Range of measurement force/moment? Signal bandwidth? Moreover, how was the down-sampling performed for the output signal originating from each force plate (that is, from 200 Hz to 50 Hz)? How was the synchronization achieved between the data originated from the positional markers, force-plate and Mexo? Why is it necessary to apply a 2nd order low-pass filter centered at 5 Hz?
3) Lines 113 – 124: The model and manufacturer information of the EMG electrodes is missing in the written text. In the filtering stage, as described within the text, authors first band-pass filtered the EMG signals (resulting bandwidth: 10 – 400 Hz), followed by high-pass filtering (resulting bandwidth: 30 – 400 Hz) and finally low-pass filtering at 2.5 Hz (resulting bandwidth: 0 Hz). How is this possible if the lower frequency component contained on the EMG signal was 30 Hz previous to the final low-pass filtering stage? It seems to me that this final stage results in complete loss of the spectral contents of the EMG signal. Or there is some information missing in the written text relative to other intermediate processing stages referred by the authors as: “filtered to remove the electronic noise” and “full-wave rectified”. If so, a thoroughly description of the entire processing routine applied to the EMG signals is required for clear readability. Perhaps a figure containing a block diagram of the entire sequence of the different filtering stages involved would be more helpful. Finally, what is the maximal voluntary contraction metric and biological substantiation?
4) Line 142: model and manufacturer information about the XoTrunk is missing.
5) Section 2.2: can the authors provide an image taken during the human trial to better illustrate the different work-related tasks? With so many lines of descriptive text devoted to each task and the control strategies for exoskeleton operation, the reader may get lost in the middle. An image can better provide a full picture of the experimental setup and intervenient tasks/materials.
6) Lines 211 – 212: What was the criteria employed by the authors to select only 4 EMG sensors from the initial set of 12 pairs? Did the authors eliminate those channels yielding larger noise interference? Or this selection was purely based on the positioning of the sensors over optimal body locations, thus associated to better performance results for all volunteers?
7) Lines 256 – 262: Which optimization technique was used to minimize the objective function in J?
Author Response
Dear Reviewer 2,
Thank you for your review and fruitful comments and suggestions. The manuscript was revised according to the reviewers’ comments. Please see the attachment for the authors’ responses to your comments and the details of the revisions to the manuscript.
Bests regards,
Ali Tabasi
Corresponding author

Round 2
Reviewer 2 Report
The authors have successfully addressed all my comments.